# Pharmacological Mechanisms and Adjuvant Properties of Licorice Glycyrrhiza in Treating Gastric Cancer

**DOI:** 10.3390/molecules28196966

**Published:** 2023-10-07

**Authors:** Joanna Japhet Tibenda, Yuhua Du, Shicong Huang, Guoqing Chen, Na Ning, Wenjing Liu, Mengyi Ye, Yi Nan, Ling Yuan

**Affiliations:** 1College of Pharmacy, Ningxia Medical University, Yinchuan 750004, China; joannajaphet14@gmail.com (J.J.T.); 20210710269@nxmu.edu.cn (Y.D.); 20220710294@nxmu.edu.cn (S.H.); 20220720321@nxmu.edu.cn (G.C.); 20220720319@nxmu.edu.cn (N.N.); 2Key Laboratory of Hui Ethnic Medicine Modernization of Ministry of Education, Ningxia Medical University, Yinchuan 750004, China; 220200943@nxmu.edu.cn; 3College of Traditional Chinese Medicine, Ningxia Medical University, Yinchuan 750004, China; yexiaomao1991@163.com

**Keywords:** licorice, glycyrrhiza, gastric cancer, adjuvant chemotherapy

## Abstract

Licorice is a remarkable traditional Chinese medicine obtained from the dried root and rhizomes of the Glycyrrhiza genus, and t has been utilized in China for many centuries. It consists of more than 300 compounds that are mainly divided into triterpene saponins, flavonoids, polysaccharides, and phenolic components. The active compounds of licorice have been found to possess multiple biological activities, including antitumor, anti-inflammatory, antiviral, antimicrobial, immunoregulatory, cardioprotective, and neuroprotective functions. In addition to providing a brief overview of licorice’s adjuvant properties, this review describes and analyzes the pharmacological mechanisms by which licorice components function to treat gastric cancer. Furthermore, licorice compounds are also found to be potent adjuvant chemotherapy agents, as they can improve the quality of life of cancer patients and alleviate chemotherapy-induced adverse effects.

## 1. Introduction

Despite a drastic decline in the incidence of Helicobacter pylori infection, gastric cancer remains the fourth most lethal malignancy worldwide, with a 5-year survival rate of about 36%. Due to poor prognosis, surgery remains the first-line strategy for the treatment of gastric cancer (GC), as most patients are diagnosed with advanced-stage gastric cancer [1]. Other treatments used for gastric cancer include chemotherapy, immunotherapy, radiotherapy, and targeted therapy [2]. GC is characterized by bloating, indigestion, nausea, and heartburn, while symptoms like jaundice, ascites, hematochezia, anemia, and excessive weight loss might be observed in the advanced stage of GC [3,4].

Licorice is a perennial plant belonging to the legume (Fabaceae) family and the *Glycyrrhiza* genus, and there are more than 20 known and accepted species in the family, such as *G. glabra, G. uralensis, G. inflata, G. lepidota, G. triphylla, G. pallidiflora, G*. *echinate,*
*G. aspera,* and *G. foetida,* [5]. The licorice plant is produced and distributed in different parts of the world, mainly including several European countries (such as Italy, Spain, Germany, and France), Northern African countries (such as Morocco, and Egypt), Asian countries (such as China, India, Afghanistan, Pakistan, Uzbekistan, Iraq, and Turkey), and some parts of Russia and the United States (Figure 1). The plant is primarily used as a natural medicinal herb and a flavoring-sweetener agent. The common scientific name given to this plant is *Glycyrrhiza glabra*, which is derived from the modern Greek translation *glycyrrhiza*, where “*glukos*” and “*rhiza*” mean “sweet root” [6].

As a traditional Chinese medicine (TCM), the dried root and rhizomes are the essential medicinal part of the plant that has been utilized in China for over 1000 years as a tonifying agent for the spleen and heart Qi, and it is also used to treat several bodily dysfunctions such as fever, cough, asthma, sore throat, spasms, gastric ulcer, tonsilitis, dyspnea, arthritis, gastritis, bronchitis, and skin diseases [7,8]. This medicinal herb has three main species, namely, *Glycyrrhiza glabra, Glycyrrhiza uralensis,* and *Glycyrrhiza inflata,* that have been registered in the latest 10th edition of the pharmacopoeia of the People’s Republic of China (2015 Edition) [9]. Apart from the Chinese pharmacopoeia, the licorice root has also been approved by the Food and Drug Administration agency (FDA) and the European Medicines Agency (EMA), demonstrating the global recognition of its potential medicinal activities. Additionally, licorice extracts have economic value and are used in the cosmetic industry, food production, and processing as an additive agent in beverages, candy, bubble gum, and food seasonings, serving as flavorings and sweeteners [10].

Due to the multifaceted benefits of the licorice root, several research studies have been conducted to explore the pharmacological mechanism of the herb. The herb is found to possess anticancer, antiviral, anti-inflammatory, antimicrobial, immunomodulatory, cardioprotective, and neuroprotective activities and treats respiratory and gastrointestinal diseases as well [11]. Nevertheless, this review aims to explore and highlight the pharmacological studies on the mechanism of action of licorice in gastric cancer.

## 2. Botanical Description

Licorice is a 40-inch-high perennial herbaceous plant consisting of pinnate leaves that are about 3–6 inches in length and folioles of about 9–17 in number. The flowering part of the plant is arranged in an inflorescence consisting of 0.8–1.2 cm long flowers that are pale blue-whitish to purple and an oblong pod fruit 2–3 cm in length with seeds; it also consists of a stolon root.

## 3. Chemical Composition of Licorice

More than 400 different chemical compounds have been extracted from licorice, among which licorice saponins and flavonoids are found to be more abundant than the other chemical components of licorice [12]. That being said, licorice mainly consists of saponins, flavonoids, phenolic compounds, volatile components, and essential oils (Table 1). 

### 3.1. Saponins

There are about 77 triterpene saponins that have been isolated in licorice, among which 50 oleanane-type triterpene saponins have been extracted from *Glycyrrhiza uralensis*, 38 from *Glycyrrhiza glabra*, and 13 from *Glycyrrhiza inflata* [1]. The most abundant saponin compound isolated among the three species is glycyrrhizin. Glycyrrhizin is a major active component of licorice that makes up about 10% of the total plant root. Given that this ingredient is 60 times sweeter than sugarcane, glycyrrhizin is recognized as a potent natural organic sweetener (about 50–100 times sweeter than sucrose). Glycyrrhizin, also known as 18β-glycyrrhizic acid, comprises two main metabolites, namely 18β-glycyrrhetyl-3-O-sulfate and glycyrrhetinic acid, that have potent multifaceted pharmacological actions such as antiulcer, antiviral, immunomodulation, anti-inflammatory, and hepatoprotective actions [13]. Other triterpene saponins found in licorice include licorice saponin H2 (C_42_H_62_O_16_), uralsaponin T (C_48_H_74_O_19_), glycyrrhetic acid 3-O-glucuronide (C_36_H_54_O_10_), araboglycyrrhizin (C_41_H_62_O_14_), and licorice saponin J2 (C_42_H_64_O_16_) [14].

### 3.2. Flavonoids

Flavonoids are the main primary components of licorice, followed by saponins. The yellow color of the licorice plant is attributed to the presence of flavonoids. Licorice flavonoids consist of more than 300 identified chemical compounds that are categorized into five main groups: flavanones, isoflavones, flavonols, flavones, and chalcones. Liquiritin, glabridin, isoliquiritigenin, liquiritigenin, isoliquiritin, rhamnoliquirilin, shinpterocarpin, licochalcone A, glucoliquiritin apioside, prenyllicoflavone A, shinflavanone, dehydroglyasperin C, licoflavone A, 1-metho-xyphaseolin, and licuraside are some of the active ingredients of licorice flavonoids. In addition, licorice flavonoids have several medicinal benefits such as antiulcer, antitumor, antimicrobial, anti-inflammatory, antiviral, antioxidant, hepatoprotective, antidepressive, and hypoglycemic action [15].

### 3.3. Polysaccharides

Glycyrrhiza polysaccharides (GPs) are also among the bioactive ingredients found in licorice. GPs are made up of several monosaccharides like mannose, galactose, glucose, and arabinose [16]. It is found that licorice polysaccharides extracted from the seeds and leaves possess higher concentrations of arabinose, galactose, mannose, and rhamnose, while the licorice polysaccharides from the roots have larger concentrations of glucose, galactose, and arabinose [17]. GPs are widely known for their immunoregulation properties, which are highly valued for their low toxicity and immunostimulant abilities. Further biological activities of GP include anticancer [18], antidiabetic [19], antioxidant [20], and antiviral [21] properties.

### 3.4. Phenolic Compounds

The phenolic components of licorice mainly consist of isoprenoid-substituted flavonoids, dihydrostilbenes, chromenes, benzofurans, coumarins, and dihydrophenanthrenes. The major active ingredients of phenols in licorice include the glycosides of isoliquiritigenin and liquiritigenin, such as liquiritin, liquiritin apioside, licuraside, and isoliquiritin. Other phenolic compounds found in licorice include licoricidin, isoangustone A, licoriphenone, licoisoflavone, gancaonin I, kanzonol R, semilicoisoflavone B, Glyinflanin G, 1-Methoxyficifolinol, 8-(gamma,gamma-Dimethylallyl)wighteone glycycoumarin, hedysarimcoumestan B, and isolicopyranocoumarin. The phenolic composition of licorice contributes to several biological functions such as antioxidant [22], antitumor [23], antithrombotic [24], antiviral [25], antibacterial [26], and antidiabetic [27] properties, to mention a few.

### 3.5. Volatile Compounds

M. Farag et al. investigated the composition of volatile compounds in the three main species of licorice (*G. glabra, G. uralensis,* and *G. inflata*) and found 38 volatile compounds, of which aldehydes and alcohol were discovered to be the major volatiles that are present in the three species [28]. Since then, more scientists have carried out additional research and have identified other volatile chemicals in licorice [29]. 5-methyl-furfural, cumin aldehyde, α-terpineol, terpinen-4-ol, carvacrol, eugenol, β-caryophyllene, p-vinyl-guaiacol, γ-terpinene, thymol, (E)-2-heptenal, 1-hexanol, benzaldehyde, (4E)-decenal, piperitone, γ -nonalactone, methyl eugenol, and β-caryophyllene oxide are some of the volatile compounds that are present in licorice. The volatile compounds of licorice may possess a potential application in medicine. For instance, compounds like thymol and carvacrol are reported in several studies to possess multiple biological properties such as antioxidant, antiviral, antibacterial, antifungal, anti-inflammatory, antispasmodic, antitumor, cardioprotective, neuroprotective, and immunomodulation properties [30,31,32,33,34,35,36,37,38].

### 3.6. Others

Licorice also consists of essential oil, from which several compounds can be isolated, such as benzoic acid, furfuraldehyde, propionic acid, 2,3-butanediol, trimethylpyrazine, 1-methyl-2-formylpyrrole, furfuryl formate, methyl ethyl ketone, ethyl linoleate, and malto [8]; organic acids like acetic, butyric, propanoic, fumaric, tartaric, malic, and citric acids [39]; and alkaloids such as 5,6,7,8-tetrahydro-4-methylquinoline and 5,6,7,8-tetrahydro-2,4-dimethylquinoline. Licorice infusions also contain other substances like protein, carbohydrates, amino acids (such as tyrosine, leucine, threonine, histidine, serine, glutamic, valine, phenylalanine, lysine, glycine, prolinealanine, and isoleucine) aspartic trace elements (such as sodium, copper, phosphorus, zinc, calcium, and potassium) fat, and silica.

## 4. The Pharmacological Mechanism of Licorice in Gastric Cancer

### 4.1. Suppression of Cellular Proliferation in Gastric Cancer

Licoricidin (LCA), an isoflavonoid extracted from *G. glabra*, was recently explored and found to possess antiproliferative properties against the MGC-803 cell line. LCA was observed to inhibit the number of colony formation in a dose-dependent manner. LCA can downregulate the expression of *cyclin d1* and *cdk4*, genes that are responsible for the progression of the cell cycle and proliferation mediation, leading to cell cycle arrest and thus the inhibition of cellular proliferation [40]. A triterpenoid saponin from *G. glabra*, glycyrrhizinic acid (GA), which is also identified as the main active component of licorice, demonstrated its antiproliferative ability in three human gastric cancer cell lines (MGC-803, BGC-823, and SGC-7901). GA inhibited cellular proliferation in the three cell lines and reduced the number of colony formations in the MGC-803 cells. GA administration significantly reduces the expression levels of cell-cycle-related proteins such as cyclin D1, D2, D3, E1, and E2, contributing to cellular proliferation inhibition [41]. Licoflavone A (LA), a flavonoid of glycyrrhiza, is also demonstrated to exert antiproliferation properties that were observed in three different gastric cancer cell lines, namely MKN-45, SGC-7901, MGC-803, and VEGF-stimulated MKN-45 cells. Several other active components of licorice, such as isoliquiritigenin, 18β-glycyrrhetinic acid, quercetin, and licochalcone A, are also reported to possess antiproliferative abilities against tumors in several other human gastric cancer cell lines like MKN28, MKN-45, SGC-7901, EBV (+) SNU719, EBV (−) MKN74, AGS, and BGC (Table 2).

### 4.2. Apoptosis Induction in Gastric Cancer Cells

Apoptosis has become one of the key areas of focus for cancer therapy. Disruption or lack of apoptosis regulation may aid in cancer cell survival, tumor progression, and even uncontrollable cellular division, which may be a threat to cancer patients. Most of licorice’s active compounds have been reported to promote apoptosis in gastric cancer cells. Through intrinsic-caspase-independent apoptosis, GA induced apoptotic cell death by upregulating the expression levels of Bax, cleaved PARP, and procaspase-3, -8, and -9 [41]. Pro-apoptotic proteins such as Bax tend to promote the release of cytochrome-c from the mitochondria to the cytosol where it may bind with *Apaf-1* and induce apoptosis. The activation of *Apaf-1* leads to apoptosome formation, which in turn activates the procaspase-9 protein and mobilizes downstream effector caspases (such as caspase-3, 7, and 8) resulting in cell death through the cleavage of several cellular substrates such as PARP. Quercetin administration was also found to promote apoptosis in nearly the same manner via p53-dependent apoptosis by upregulating the levels of PUMA, Bax, cleaved forms of PARP, and caspase-3 and -9 in EBV (−) human gastric cancer cells (MKN74) [42]. Licochalcone A is reported to induce apoptosis as well, in a dose-dependent manner, by regulating the expression quantity of PARP, caspase-3, Bcl-2, and Bax proteins in AGS, MKN-28, and MKN-45 gastric cancer cells [43]. LA promoted apoptosis in VEGF-stimulated MKN-45 cells by regulating the mitochondrial pathway, whereas LA increased the expression of Bax/Bcl-2 protein, which in turn activated downstream of apoptosis promoters, including Cyt C, caspase-9, and cleaved caspase-3. Moreover, apoptotic bodies, nuclear deformation, and shrinkage were observed upon LA treatment. Isoliquiritigenin (ISL) downregulated the levels of Bcl-2 while increasing the expression of Bax and caspase-3 in the MKN28 cell line, leading to apoptotic cell death [44]. In addition, the expression levels and ratios of Beclin 1 and LC3II/LC3I were found to be elevated after the administration of ISL, whereas the p62 protein was found to be downregulated. This suggests that ISL may also have autophagy modulation functions.

### 4.3. Inhibition of Cellular Invasion and Metastasis in Gastric Cancer

One of the main hindrances to tumor suppression and cancer treatments is the tumor’s ability to become invasive and metastatic. The invasive and metastatic properties of the tumor contribute highly to the deterioration of the quality of life of cancer patients and are also marked as the most common cause of death in cancer patients [45]. Cai H et al. illustrated the mechanism of action that 18β-glycyrrhetinic acid (18β-GA) may exert to suppress the invasive and metastatic properties of the human gastric cancer SGC-7901 cells. According to their research, 18β-GA could downregulate intracellular ROS production, vimentin, MMP-2 and -9 expression while increasing E-cadherin levels, which are all vital for tumor metastasis and progression, as they are found to be linked in the EMT process (Figure 2). Licoflavone A (LA) was also found to inhibit the EMT process in VEGF-stimulated MKN-45 cells by suppressing proteins involved in EMT, including MMP2, MMP9, and N-cadherin, whereas E-cadherin expression was increased, therefore showing that LA can suppress invasion and migration of gastric cancer cells. Other compounds of licorice have also demonstrated their antimetastatic ability in nearly the same manner, including LCA and ISL; their action was demonstrated in human gastric carcinoma MGC-803 and MKN28 cells mainly via suppressing EMT-associated proteins and the PI3K/AKT/mTOR pathway, respectively.

### 4.4. Regulation of microRNAs

Modulating microRNA expressions has become a promising cancer treatment approach, as they are found to be involved in angiogenesis, proliferation, metastasis, and apoptosis of cancer cells [46,47].

Li Xia et al. demonstrated the role of 18β-GRA in regulating microRNAs to alleviate gastric cancer. 18β-GRA was able to upregulate intracellular miR-345-5p levels, which in turn targeted the expression of TGM2 levels, a protein involved in cellular matrix adhesion that is known to aid cancer cells’ survival, motility, and metastasis [48]. The upregulation and downregulation of miR-345-5p and TGM2 expressions, respectively, led to the apoptotic cell death and cell cycle arrest of the AGS and HGC-27 cells [49]. Moreover, 18β-GA was also demonstrated to mediate the miR-149-3p-Wnt-1 signaling pathway via decreasing COX-2 and Wnt-1 levels while increasing miR-149-3p to induce anticancer effects in MKN-1 and BGC-823 cells [50].

### 4.5. Regulation of Molecular Signaling Pathways

LCA could induce multiple signaling pathways to exert its anticancer effects. Hao W et al. explored the role of LCA on PI3K/AKT/mTOR and MAPK signaling pathways in treating gastric cancer and found that LCA could inhibit cellular proliferation, induce ROS production, induce apoptotic cell death, and suppress tumor growth in vivo by regulating the two pathways. LA also downregulated the PI3K/AKT and MEK/ERK signaling pathways by targeting the VEGFR-2 protein to inhibit cellular proliferation, invasion, migration, and EMT, inducing cell cycle arrest and apoptosis. LCD was found to regulate the Isoprenyl carboxyl methyltransferase (ICMT)/ RAS pathway to induce its anticancer effects in MGC-803 cells. LCD suppressed ICMT expression, which is known to enhance tumor growth and survival in several cancers [51]. ICMT inhibition led to the blocking of Ras/Raf/Mek/ERK signaling, resulting in the potent anticancer effects of LCD. GA modulated the PI3K/AKT pathway to induce cell cycle arrest and apoptotic death in MGC-803 cells. 18β-GA regulated the ROS/PKC-α/ERK pathway to induce antimetastatic and anti-invasive properties of GC cells by suppressing ERK phosphorylation, ROS production and PKC-α levels.

### 4.6. Immunoregulatory Functions

The immune system plays a major role in the eradication of cancer development. Treatments like immunotherapy alone or in combination with other chemotherapy drugs have been used to boost the immunity of cancer patients and destroy cancer cells [52].

Recently, a novel compound of licorice glycyrrhiza named licorice polysaccharide (GPS-1) was found to possess immunomodulatory functions. GPS-1 elevated the expression ratio of CD3+CD8+ and CD3+CD4+ T lymphocytes, promoted the maturation and phagocytosis of dendritic cells (DCs), and enhanced the cytokine production of IL-4 and IFN-γ [53]. Studies have shown that glycyrrhizic (GL) and glycyrrhetinic acid (GA) can modulate multiple components that are involved in immunoregulation and inflammation, such as TNF-α, IL-13, IL-12, IL-10, IL-6, IL-3, IL-4, IL-5, IL-1β and eotaxin secretion [54]. GL can enhance the maturation and activities of DCs by elevating the levels of MHC-II, CD40, and CD86, leading to increased t-cell proliferation and cytokine secretion of IFN-γ and IL-10, while IL-4 secretion is reduced [55]. GL and ISL can regulate innate immunity via TLR4/MD-2 signaling by inhibiting TNF-α and IL-6 expression and suppressing the activation of NF-κB and MAPKs (such as p38, ERK, and JNK) [56]. Furthermore, the effects of licorice polysaccharides in CT-26 tumor-bearing BALB/c mice were observed and found to be closely related to their immunomodulatory properties. L. polysaccharides could inhibit tumor growth as a result of CD4+ and CD8+ activation; the upregulation of tumor-suppressing cytokines, including IL 2, IL 6, and IL 7; and the downregulation of TNFα [57].

## 5. Pharmacological Mechanisms of Licorice in Conjunction with Other Drugs against Gastric Cancer

### 5.1. Combined with Chemotherapy Medications against Gastric Cancer

Chemotherapy resistance has become a worldwide epidemic, contributing to the backsliding of cancer eradication. Cisplatin (DDP) is one of the main chemotherapy treatments used for the treatment of gastric cancer. And like other chemotherapy drugs, DDP was also reported to be a victim of drug resistance [58,59]. Wei Feng et al. explored and revealed that a combination treatment of liquiritin with DDP could alleviate DDP resistance in human gastric cancer [60]. The combination of the two had potent anticancer effects compared to when the two drugs were administered separately. Furthermore, 5-FU was used to enhance the anticancer effects of LCA in SGC7901 and MKN-45 cells. The combination of 5-FU with LCA was shown to inhibit the proliferation of gastric cancer cells by promoting cell cycle arrest and apoptotic death.

### 5.2. Combined with Other Compounds or Drugs against Gastric Cancer

Recently, the synergistic impact between the tumor necrosis factor-related apoptosis-inducing ligand (TRAIL) and LIQ against gastric cancer was explored. TRAIL is a cytokine from the TNF family, and it is known for its ability to impede tumor progression by promoting apoptotic death without damaging healthy cells [61]. Xie Rui et al. investigated the combinational treatment between the two by using TRAIL-resistant GC cell lines, and the study demonstrated that TRAIL+LIQ could significantly inhibit gastric cancer progression by suppressing cellular proliferation, migration, and tumor growth in vivo by regulating ROS and JNK activation [62].

**Table 2 molecules-28-06966-t002:** Pharmacological studies on the mechanisms of action of licorice in gastric cancer.

Active Components of Licorice	Experimental Model	Mechanism of Action	Signaling Pathways Involved	Journal Citation
Licoricidin (LCD)	In vitro: Human gastric cancer cell line of MGC-803 In vivo: Male nude mice 5 weeks old, 20 ± 2 g four groups (*n* = 6) Administration: Dosage; 10 mg/kg of LCD, 20 mg/kg of LCD, 20 mg/kg of 5-FU Route; Subcutaneous	Inhibited cellular proliferation, cellular migration, and invasion, induced apoptosis and cell cycle arrest at G0/G1 phase. Inhibited tumor growth.	Isoprenyl carboxyl methyltransferase (ICMT)/RAS pathway	[40]
Glycyrrhizic acid (GA)	Human gastric cancer cell line of MGC-803, BGC-823, and SGC-7901.	Inhibited cellular proliferation, promoted cell cycle arrest at G1/S-phase by ↓ cyclin D1, D2, D3, E1, and E2. Induced apoptosis by ↑ levels of Bax, cleaved PARP, and procaspase-3, -8, -9.	PI3K/AKT pathway	[41]
18β-glycyrrhetinic acid (GRA)	In vitro: Human gastric cancer cell line of MKN-1, and BGC-823 In vivo: Male transgenic mice 6-week-old, two groups (*n* = 40) Administration: Dosage; distilled water containing 0.05% GRA Route; Oral	Inhibited cellular proliferation, induced cell cycle arrest, and apoptosis. Inhibited tumor growth	miR-149-3p-Wnt-1 signaling	[50]
Liquiritin (LIQ) + Cisplatin (DDP)	Human gastric cancer cell line of SGC7901/DDP In vivo: male BALB/c-nu mice 5-week-old, 15–18 g four groups (*n* = 10) Administration: Dosage; 15 mg/kg of LIQ, 3 mg/kg of DDP Route; Intraperitoneal injection	LIQ relatively inhibited the proliferation and migration of DDP-resistant gastric cancer cells. DDP+LIQ promoted cell cycle arrest at G0/G1 by ↓ cyclin D1, cyclin A, and ↑ CDK4 and p53 and p21. DDP+LIQ induced apoptosis and autophagy. Inhibited tumor growth of xenograft mice.		[60]
Licoflavone A (LA)	In vitro: Human gastric cancer cell line of SGC-7901, MKN-45, MGC-803and VEGF-stimulated MKN-45 cells. In vivo: Male BALB/c-nude mice 4–6-week-old, 18 ± 2 g Administration: Dosage; 50 mg/kg of LA Route; Oral	Suppressed cellular proliferation. Induced apoptosis and cell cycle arrest at G1 phase, Inhibited the migration, invasion, and EMT of VEGF-stimulated MKN-45 cells. Inhibited tumor growth.	PI3K/AKT and MEK/ERK signaling pathways.	[63]
Isoliquiritigenin (ISL)	In vitro: Human gastric cancer cell line of MKN28	Inhibited cellular proliferation, migration, and invasion. Promoted apoptosis and autophagy	PI3K/AKT/mTOR	[44]
18β-glycyrrhetinic acid (18β-GA)	In vitro: Human gastric cancer cell line of SGC-7901	Inhibited cellular proliferation, migration, and invasion. ↓ ROS formation, and expression of MMP-2 and 9, PKC-α, ERK, and vimentin.	ROS/PKC-α/ERK pathway	[64]
Quercetin (QC)	In vivo: Human gastric cancer cell line of EBV (+) SNU719, EBV (−) MKN74 Female NOD/SCID mice five weeks old, two groups (*n* = 15) Administration: Dosage; 30 mg/kg of QC Route; Oral	Inhibited tumor growth of the xenograft mice. Suppressed EBV viral proteins expression; (EBNA-1 and LMP-2) Promoted p53-dependent apoptosis by increasing the expression of caspase-3, -9, and Parp.		[42]
Licochalcone A (LCA) + 5-fluorouracil (5-FU)	In vitro: Human gastric cancer cell line of SGC7901 and MKN-45	LCA suppressed cellular proliferation, induced apoptosis, and cell cycle arrest at G2/M transition. LCA+5-FU enhanced the anticancer effects.		[65]
Liquiritin (LIQ) + TRAIL	In vitro: Human gastric cancer cell line of AGS and SNU-216. In vivo: Male BALB/c-nu mice 5 weeks old,15–18 g Administration: Dosage; 20 mg/kg of LIQ, 100 mg/mouse of TRAIL Route; Intraperitoneal	Suppressed cellular proliferation, and migration. Induced apoptosis both in vitro and in vivo, enhanced activation of ROS and JNK. Inhibited tumor growth in vivo.		[62]
Licochalcone A	In vitro: Human gastric cancer cell line of AGS, MKN-28, and MKN-45.	Inhibited cellular proliferation. Promoted cell cycle arrest at the G2/M transition by ↓ levels of cyclin A, B, and MDM2 and ↑ Rb expression. Induced apoptosis by regulating PARP, caspase-3, Bcl-2 and Bax expressions.		[43]
Glycyrrhetinic acid (GA) 11-deoxy glycyrrhetinic acid (11-DOGA)	In vitro: Human gastric cancer cell line of BGC823 and SGC7901. In vivo: Nude Mice Administration: Dosage; 0, 10, 20, and 30 mg/kg of GA, 0, 10, 20, and 30 mg/kg of 11-DOGA Route; Subcutaneous injection	Suppressed cellular proliferation. Promoted cell cycle arrest in G2 Phase by ↑ p21 expression and ↓ cdc2 and cyclin B1. Induced apoptosis by ↓ Bid expression and activated PARP cleavage. Inhibited tumor growth in vivo.	Bid-mediated mitochondrial pathway.	[66]
Licochalcone A	In vitro: Human gastric cancer cell line of BGC. In vivo: SPF KM mice, 6–8 weeks, 13–15 g, two groups (*n* = 10). Administration: Dosage; 200 and 400 μM of LA Route; Intratumoral injection	Inhibited cell proliferation, and induced apoptosis. Inhibited tumor growth in vivo.	PI3K/AKT and ROS-mediated MAPK signaling pathway	[67]

## 6. Toxicology Studies

Research studies have reported *Glycyrrhiza glabra* to be mildly toxic and can induce significant adverse effects such as hypertension, gastrointestinal symptoms, and neurotoxicity [68]. The toxicity of *G. glabra* has been found to depend on the mode of administration. Oral administration is regarded as the safest compared to other means of administration like IP or IV due to the first-pass effect and decreased absorption following oral consumption [69]. Additionally, licorice compounds have been shown to have a significant cytotoxic effect against gastric cancer cell lines (Table 3). Some studies compared the cytotoxicity of licorice in normal gastric and gastric cancer cells; the results revealed the cytotoxicity in gastric cancer cells to be higher than in normal cells when administered with licorice compounds. This suggests that licorice components could be beneficial as adjuvant therapy in cancer treatment.

## 7. Discussion

Licorice is an important traditional herb that is widely utilized in nutritional supplements and is a highly prescribed ingredient in TCM treatment. Despite licorice being commonly used as a flavoring agent and in dietary supplements for multiple bodily disorders, it has a lesser impact on being utilized as a chemopreventive agent. There are fewer clinical trial studies to support the effect of licorice on cancer, especially gastric cancer. Nevertheless, several clinical studies demonstrate the safety and adjuvant properties of licorice in cancer: a combinational therapy consisting of an oxaliplatin regimen and traditional medicines (TM) (including *Glycyrrhiza uralensis, Astragalus membranaceus, Atractylodes macrocephala, Poria cocos, Coix lacryma-jobi,* and *Panax ginseng*) was shown to inhibit nausea and vomiting induced by chemotherapy. The combinational treatment of the two was also able to enhance gastroprotective effects as well as regulate antioxidant effects and gastrointestinal motility [70]. Terminal cancer patients diagnosed with lung, liver, colorectal, stomach, and other types of cancer were kept under palliative care and given a traditional herb diet consisting of peony and licorice root. After 10 days, their pain levels were monitored, and the group that received the herb diet had increased pain relief compared to the group that received a conventional hospital diet [71]. Recent randomized clinical trials demonstrated licorice’s ability to alleviate pain and radiotherapy-induced effects like oral mucositis by acting as a mucoadhesive film [72,73]. The mucopreventive effects of licorice may be attributed to the anti-inflammatory properties of the herb through the scavenging of free radicals to prevent the formation of reactive oxygen species and the downregulation of proinflammatory cytokines and prostaglandin E2 secretion [74,75]. In addition, several herbal formulations containing licorice components such as TJ41 (consisting of *Glycyrrhiza radix, Pinellia tuber, Scutellaria baicalensis, Zingiberis rhizoma, Zizyphi fructus, Panax ginseng,* and *Coptidis rhizoma*), PHY 906 (consisting of *Glycyrrhiza uralensis, Scutellaria baicalensis Georgi, Paeonia lactiflora Pall,* and *Ziziphus jujuba Mill*), TJ84 (consisting of rhubarb and glycyrrhiza), TJ48 (consisting of *Ginseng radix, Glycyrrhizae radix, Astragali radix, Rehmanniae radix, Angelicae radix, Cinnamomi cortex, Poriacocos, Atractylodis lanceae rhizoma, Paeoniaeradix,* and *Ligustici hizome*), and TJ43 (consisting of licorice root, *Ginseng Radix, Poria cocos, Rhizoma atractylodis macrocephalae, pinelliae tuber, pericarpium citri,* jujube, and ginger) are reported to have antitumor effects and enhance the quality of life of cancer patients by mitigating chemotherapy-induced effects such as GI complications, fatigue, anemia, appetite, and mucositis [76,77,78,79,80].

Moreover, isoliquiritigenin was found to enhance the anticancer effects of 5-FU and attenuate chemoresistance induced by gastric cancer cell stemness and tumor microenvironment [81]. ISL inhibited the expression of *GRP78*, a gene associated with reducing the effectiveness of anticancer drugs and promoting chemoresistance. Glycyrrhizin sensitized cancer cells to radiation and cisplatin treatment by regulating a protein known for its involvement in tumor metastasis and proliferation, the high mobility group protein B1 (HMGB1) [82]. Glycyrrhetinic acid reduced the pulmonary injury induced by radiation therapy by targeting the TGF-β1/Smad signaling pathway [83]. The polytherapy treatment consisting of glycyrrhizin and lamivudine could attenuate cisplatin resistance by downregulating multidrug resistance proteins such as MRP2, -3, and -5 [84]. These studies demonstrate that licorice has significant synergistic effects with chemotherapy treatments. Chemotherapy treatments were more effective when licorice compounds were added to the regimen. Licorice amplifies the effectiveness of chemotherapy agents by sensitizing cancer cells to them and alleviating the side effects that can be accompanied by chemotherapy. This suggests that licorice’s compounds may not only confer protection against chemotherapy-induced side effects but also chemoresistance, which contributes highly to the backsliding of cancer treatment through cancer recurrence.

## 8. Conclusions and Prospects

The components of licorice are mainly categorized into saponins, flavonoids, phenolic compounds, volatile components, and essential oils. Among these, a triterpenoid saponin known as glycyrrhizinic acid (GA) is regarded as the main chemical compound of licorice. The major active components of licorice exert their mechanism of action against gastric cancer by mainly inducing their antimetastatic properties, leading to apoptosis and tumor suppression via regulating several signaling pathways involved in cellular growth and development. Moreover, this article has highlighted the immunoregulation activities of licorice, which are essential in cancer eradication; however, the immunomodulatory functions of licorice in gastric cancer have not yet been explored as they have been in the colon [18,85], lung [86,87], and breast cancer [88]. Therefore, we employ scientists to further investigate and analyze the role of glycyrrhiza components in treating gastric cancer, especially their immunoregulation properties. In conclusion, this review has highlighted the anticancer effects of licorice in gastric cancer and revealed the adjuvant properties of licorice components, which might aid and encourage more scientists to further research the novel pharmacological mechanisms of licorice in gastric cancer, as there are still not enough research studies to back up the potent anticancer properties of licorice and its diverse components.

## Figures and Tables

**Figure 1 molecules-28-06966-f001:**
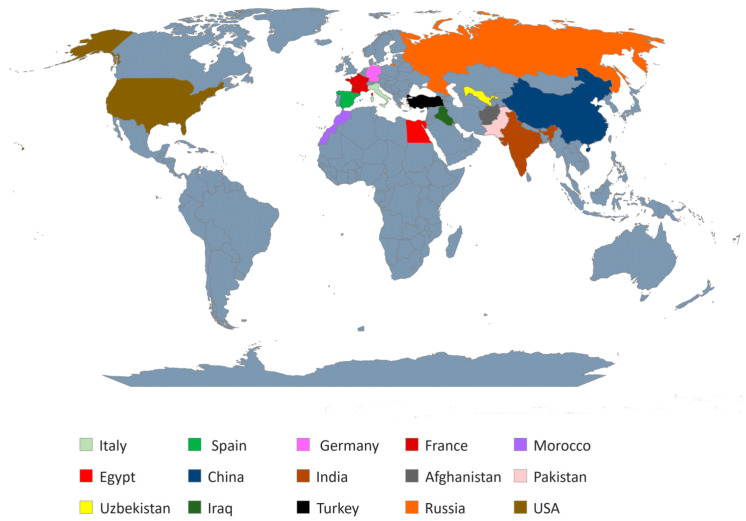
A world map displaying some of the major licorice-producing countries.

**Figure 2 molecules-28-06966-f002:**
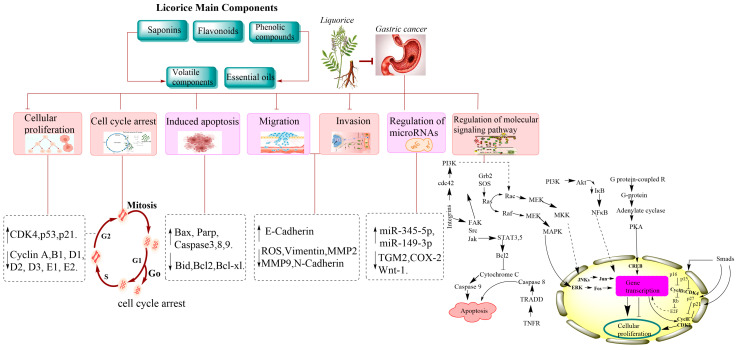
A schematic drawing illustrating the pharmacological mechanism of licorice in gastric cancer.

**Table 1 molecules-28-06966-t001:** Chemical structures and categories of licorice glycyrrhiza active components.

Compound	Chemical Formula	Chemical Structure	Category
Glycyrrhizin	C_42_H_62_O_16_	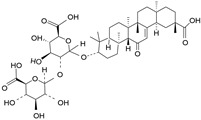	Triterpene saponin
Glycyrrhetinic acid	C_30_H_46_O_4_	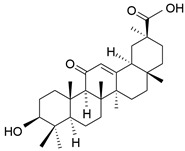	Triterpene saponin
18β-glycyrrhetyl-3-O-sulfate	C_30_H_46_O_7_S	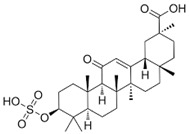	Triterpene saponin
Liquiritin	C_21_H_22_O_4_	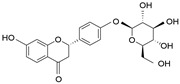	Triterpene saponin
Licochalcone A	C_21_H_22_O_4_	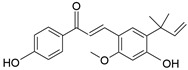	Flavonoid
Glabridin	C_20_H_20_O_4_	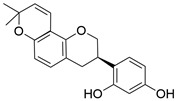	Flavonoid
Isoliquiritigenin	C_15_H_12_O_4_	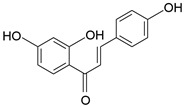	Flavonoid
Liquiritin apioside	C_26_H_30_O_13_	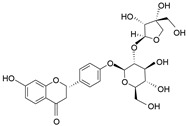	Flavonoid
Liquiritigenin	C_15_H_12_O_4_	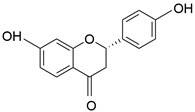	Flavonoid
Isoliquiritin	C_21_H_22_O_9_	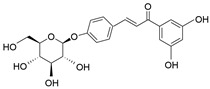	Flavonoid
Licoriphenone	C_21_H_24_O_6_	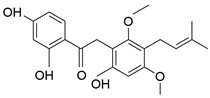	Phenol
Kanzonol R	C_22_H_26_O_5_	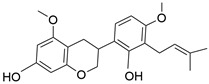	Phenol

**Table 3 molecules-28-06966-t003:** Cytotoxicity of licorice compounds in gastric cell lines and normal gastric cell line.

Compounds	Cell Lines	Dosage	Cytotoxic Outcome/IC50 (μM)
Licochalcone A	GES-1 AGS MKN-28 MKN-45	0, 10, 25, 50 and 100 µM, 48 h	92.7 41.1 42.0 83.7
Licoricidin (LCD)	MCG-803	1.5625, 3.125, 6.25, 12.5, 25, 50, 100, and 200 μM, 24 h	10.41
Glycyrrhizic acid	MGC-803 BGC-823 SGC-7901	0, 0.5, 1, 2, 3, 4 mg/mL, 48 h	≈2 mg/mL
Licoflavone A (LA)	GES-1 SGC-7901 MKN-45 MGC-803	0, 6.25, 12.5, 25, 50, and 100 μM, 24 h	180.30 78.08 43.26 124.50
18β-glycyrrhetinic acid (18β-GA)	SGC-7901 cells	0, 20, 40, 60, 80, 100 and 120 μM, 24 h	cytotoxicity was observed at a concentration > 80 μM
Liquiritin (LIQ)	GES-1 AGS SNU-216	0, 25, 50, 100, 150 and 200 uM, 24 h	cytotoxicity > 150 μM 185.73 198.86

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
