# Peer review of "Pharmacological Mechanisms and Adjuvant Properties of Licorice Glycyrrhiza in Treating Gastric Cancer"

_molecules, 2023, doi:10.3390/molecules28196966_

Round 1

Reviewer 1 Report

This is a comprehensive review on the effect of Licorice in gastric cancer treatment. Overall, the information is well presented. I have several suggestions:

1.     Figure 1 is not necessary and does not add much information to the readers.

2.     The authors should consider to add more details in figure 2. For each effect of licorice, which pathway and genes are affected?  

English is ok.

Reviewer 2 Report

In this review, authors describe and analyze the pharmacological mechanisms by which licorice components function to treat gastric cancer. Also the pharmacological mechanism of licorice in gastric cancer are analyzed. The topic of the work is excellent, but there are concerns regarding the presentation of the work that the authors need to pay attention to:

 General Comments:

1 - The review has similarities with the following article in terms of the subject and some cases, so it is necessary for the authors to pay more attention to the differences and reviews of similar work. Also, figure 1 of the manuscript is mentioned as figure 1 of the article, but why are the mentioned countries not the same? Please explain more about these uncertainties:

Wahab, S. et. al.,Glycyrrhiza glabra (Licorice): A Comprehensive Review on Its Phytochemistry, Biological Activities,Clinical Evidence and Toxicology. Plants 10, 2751 (2021), 10, 2751. https://

doi.org/10.3390/plants10122751

 2 - What is the importance of this research compared to previous works? What does it add to previous research?

3 - It is better to make a comparison regarding the effectiveness of licorice compared to chemotherapy agents or similar supplements

 4- Has there been a study about licorice in radiation protection and the effects of radiation during cancer radiation therapy?

 5-Some sections do not have references? Please check. Like sections: 4.2 and 5.2 ...

Reviewer 3 Report

This review by Tibenda et al. discussed about pharmacological mechanisms and adjuvant properties of licorice glycyrrhiza in Treating Gastric Cancer. The manuscript is containing useful information about licorice’s properties and pharmacological mechanisms to treat gastric cancer. This review may be considered for publication in molecules journal after below correction.

The chemical structures in table 1 are cut off in column 3. These should be fully visible. Please correct structure of Kanzonol R as there should not be methyl on oxygen of the ring.

Round 2

Reviewer 2 Report

Thanks to the authors for answering my doubts and editing the article.